# Variational Component Decoder for Source Extraction from Nonlinear Mixture

## Abstract

In many practical scenarios of signal extraction from a nonlinear mixture, only one (signal) source is intended to be extracted. However, modern methods involving Blind Source Separation are inefficient for this task since they are designed to recover all sources in the mixture. In this paper, we propose supervised Variational Component Decoder (sVCD) as a method dedicated to extracting a single source from nonlinear mixture. sVCD leverages the sequence-to-sequence (Seq2Seq) translation ability of a specially designed neural network to approximate a nonlinear inverse of the mixture process, assisted by priors of the interested source. In order to maintain the robustness in the face of real-life samples, sVCD combines Seq2Seq with variational inference to form a deep generative model, and it is trained by optimizing a variant of variational bound on the data likelihood concerning only the interested source. We demonstrate that sVCD has superior performance on nonlinear source extraction over a state-of-the-art method on diverse datasets, including artificially generated sequences, radio frequency (RF) sensing data, and electroencephalogram (EEG) results.

## 1 Introduction

Signal extraction from nonlinear mixture is a recurring yet hard research problem in signal processing and representation learning. To tackle this problem, conventional methods can be divided into two categories. One is Blind Source Separation (BSS) that recovers all sources in the mixture (Comon & Jutten, 2010), the other is to leverage available prior knowledge to extract only the desired components from the mixture (Leong et al., 2008). Following the hardness results of nonlinear BSS based on nonlinear Independent Component Analysis (ICA) (Hyvärinen & Pajunen, 1999), many attempts have been made to realize nonlinear ICA and to further connect it with representation learning (Hyvärinen & Morioka, 2016; 2017; Khemakhem et al., 2020). However, in many practical scenarios, only one or few sources are supposed to be retrieved for the concerned application; for example, heartbeat monitoring, especially that achieved by contact-free approaches (Ha et al., 2020), demands the heartbeat signal (waveform) to be extracted out of the nonlinear mixture with other signals such as body movements and respiration. Compared with nonlinear BSS, *nonlinear source extraction* can be far more efficient and effective under these scenarios: on one hand, extracting only the source of interest could significantly reduce the computation cost for the same category of algorithms (e.g., deep-learning based signal extraction would be computationally cheaper than deep-learning based signal separation). On the other hand, if prior knowledge of the source is available, the training may substantially leverage such knowledge so as to sufficiently improve the (later) inference accuracy.

Recently, there is a rising interest in investigating disentangled representation learning exploiting deep learning techniques. Different from conventional source separation methods that demand hard-coded rules, a deep learning model aims to learn the underlying generating factors automatically and adaptively. To this end, the frameworks of variational auto-encoder (VAE) (Kingma & Welling, 2013) and generative adversarial network (GAN) (Goodfellow et al., 2014) are widely used, and variants modifying designs and training techniques to enhance disentanglement have also been proposed, such as $\beta$-VAE (Higgins et al., 2016). However, as the existing proposals mostly focus on disentangling on the representation in image or video data (Kingma & Welling, 2013; Higgins et al., 2016; Fabius & Van Amersfoort, 2014; Yingzhen & Mandt, 2018) where signal mixture never take place, we cannot directly apply the idea of disentangled representation learning to our problem of

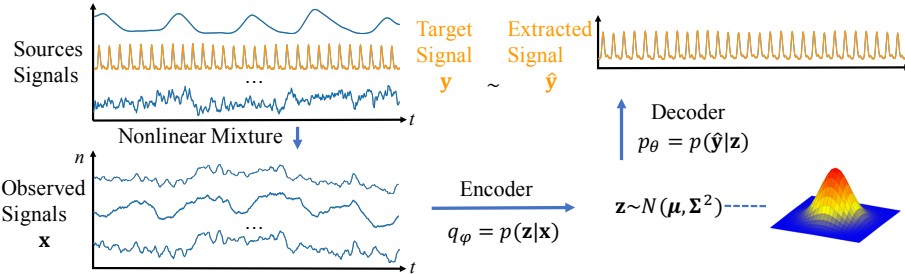

Figure 1: Source extraction from nonlinear mixture with sVCD.

source extraction (a special form of disentanglement) from nonlinear time-series mixture. Though a few results have been obtained for disentangling the sources in nonlinear mixing using deep learning, they fall into the category of BSS (Hyvärinen & Morioka, 2016; Khemakhem et al., 2020).

In this paper, we propose *supervised Variational Component Decoder* (sVCD) as a robust deep-learning framework for source extraction from nonlinear mixture. As illustrated in Figure 1, sVCD aims to recover the interested source $\mathbf{y}$ from observed signals $\mathbf{x}$ formed by source signals mixed in a nonlinear manner. In particular, sVCD involves an encoder to analyze the temporal features of the observed signals and hence to encode them to the hidden state (posterior distributions in a latent space), and it then decodes the hidden state to recover the interested source. A modified variational inference is applied to help approximate the intractable posterior and to achieve a disentangled representation in the latent space dedicated to the interested source, guided by the prior knowledge applied to train sVCD. Essentially, while $\mathbf{y}$ is available for supervised training, sVCD takes $\mathbf{x}$ as the sole input during the testing phase to obtain $\hat{\mathbf{y}}$. To summarize, our main contributions include:

- We prove that nonlinear source extraction can be formulated as modified variational inference and achieved by maximizing a tighter variational lower bound.
- We design a novel sequence-to-sequence (Seq2Seq) translation architecture; it leverages the modified variational inference to extract the desired source from nonlinear mixture.
- We provide empirical evidence that sVCD outperforms a state-of-the-art method based on extensive evaluations under different nonlinear mixing scenarios.

The rest of the paper is organized as follows. Section 2 presents related works. Section 3 introduces the mathematical framework for source extraction from nonlinear mixture. Section 4 presents the details sVCD model. Section 5 introduces the datasets and reports the evaluation results. Finally, Section 6 concludes this paper and points out potential future applications.

## 2 RELATED WORKS

Numerous algorithms have been developed to solve the BSS problem (Comon & Jutten, 2010): they all aim to decompose signal mixtures into individual components without knowing information about the source signals or the mixing process. Among them, Non-negative Matrix Factorization (NMF) decomposes the signal by using low-rank approximations under the non-negative constraint (Wang & Zhang, 2012). Independent Component Analysis (ICA) assumes that the underlying signals are statistically independent to separate them (Hyvärinen & Oja, 2000). BSS achieved by masking and re-weighting the frequency spectrograms have also been implemented by Hidden Markov Models (HMMs) (Roweis, 2000) and segmentation methods (Bach & Jordan, 2005).

One common assumption of regular BSS is that the mixing process is linear, under which the statistical independence of the underlying sources is a sufficient condition to constrain the linear unmixing function. However, as we extend BSS to nonlinear mixture, the statistical independence of the sources can no longer be treated as a sufficient constraint for the unmixing (Hyvärinen & Pajunen, 1999). Harmeling et al. (2003) and Hyvärinen & Morioka (2016) have proposed to utilize temporal structure in the mixture to overcome this challenge. Furthermore, several properties on the mixed sources, e.g., autocorrelation (Sprekeler et al., 2014), general non-Gaussian temporal dependency (Hyvärinen & Morioka, 2017), or non-stationarity (Hyvärinen & Morioka, 2016), can also provide sufficient constraints for solving the problem.

Aother broadly related topic is disentanglement representation learning (Bengio et al., 2013). The word "disentanglement" means decoupling of generating factors. Unlike BSS performing separation in *signal space*, disentanglement representation learning uncovers the underlying factors of variation in *latent space*. With the development of deep learning, disentanglement representation learning based on VAEs has gained momentum (Kingma & Welling, 2013). By augmenting the lower bound formulation with the coefficient that regulates the independence prior, Higgins et al. (2016) proposes $\beta$-VAE, a framework to discover interpretable latent representations automatically. Subsequent works (Burgess et al., 2018; Rolinek et al., 2019; Chen et al., 2018) have led to significant accomplishments in understanding the capabilities of nonlinear disentanglement in VAEs. Researchers have also been seeking the formal definition of disentanglement in several works (Ridgeway, 2016; Higgins et al., 2018; Eastwood & Williams, 2018).

The aforementioned algorithms assume that the sources are completely unknown, however, in many practical cases, we have some prior knowledge available and do not actually have to perform source separation in a completely blind manner. As such, these algorithms can be extended to incorporate the knowledge from the priors. In Calhoun et al. (2005), paradigm information was employed to aid ICA analysis, and the results are shown to be more robust to noise and outperform basic ICA. In another work (Jeong et al., 2009), the original NMF algorithm for BSS is significantly improved by enforcing a constraint that requires disjointness under the semi-blind denoising framework. Last but not least, Kameoka et al. (2018) achieves source separation by leveraging conditional VAE (Sohn et al., 2015) with source class as labels in priors during training. All the aforementioned approaches do not seem to suit our need for extracting specific source(s).

## 3 VARIATIONAL INFERENCE FOR SIGNAL EXTRACTION

Unlike VAE (Kingma & Welling, 2013) that aims to recover input by finding an efficient encoding in latent space, our sVCD focuses on extracting a specific component from nonlinear mixture. Therefore, if we directly borrow the idea of variational inference from VAE, the variational lower bound will not be as tight. As such, we propose and prove a new lower bound for our problem setup, so as to endow a new mathematical framework for sVCD.

Let $\mathbf{x}$ be a nonlinear mixture of several independent sources. In order to find a representation of $\mathbf{x}$, Bayesian modeling (Lee, 1989) can be used to encode the beliefs about the processes that generate $\mathbf{x}$ from all components into a model $\mathcal{M}$ with latent vectors $\mathbf{z}$ and parameters $\theta_{\mathcal{M}}$. The model is then "learned" by inferring $\mathbf{z}$ and adjusting the parameters $\theta_{\mathcal{M}}$. In other words, given the observed mixture $\mathbf{x}$, Bayesian modeling aims to infer the posterior $p(\mathbf{z}|\mathbf{x})$. However, since the posterior distribution is intractable, an optimization approach, which is called variational inference (Blei et al., 2017), can be leveraged to approach the problem. To be specific, a surrogate distribution $q(\mathbf{z})$ is employed to approximate $p(\mathbf{z}|\mathbf{x})$ by minimizing their KL divergence (Van Erven & Harremos, 2014). However, the KL divergence itself still involves the intractable posterior $p(\mathbf{z}|\mathbf{x})$. To cope with this issue, the KL divergence is decomposed into:

$$\mathbb{KL}\left(q(\mathbf{z})\|p(\mathbf{z}|\mathbf{x})\right) = \log p(\mathbf{x}) - \mathbb{E}_{q(\mathbf{z})}\left[\log\frac{p(\mathbf{x},\mathbf{z})}{q(\mathbf{z})}\right]. \tag{1}$$

Since the marginal log-likelihood $\log p(\mathbf{x})$ is independent of the variational distribution $q(\mathbf{z})$, the KL divergence can be minimized by maximizing the variational lower bound $\mathbb{E}_{q(\mathbf{z})}\left[\log\frac{p(\mathbf{x},\mathbf{z})}{q(\mathbf{z})}\right]$. As a particular type of variational inference method, VAE uses deep neural networks to approximate the generative models. It should be noted that in VAE, the distribution $q$ is conditioned on the observation $\mathbf{x}$, and approximated by an encoder network $q(\mathbf{z}|\mathbf{x})$. As such, the variational lower bound of VAE can be expressed as:

$$\mathbb{E}_{q(\mathbf{z}|\mathbf{x})}\left[\log\frac{p(\mathbf{x},\mathbf{z})}{q(\mathbf{z}|\mathbf{x})}\right] = \mathbb{E}_{q(\mathbf{z}|\mathbf{x})}\left[\log p(\mathbf{x}|\mathbf{z})\right] - \mathbb{KL}\left(q(\mathbf{z}|\mathbf{x})\|p(\mathbf{z})\right), \tag{2}$$

By slight modifications (Higgins et al., 2016; Kim & Mnih, 2018), the latent representation $\mathbf{z}$ of VAE shows disentanglement properties, suggesting that a regular VAE can be used for separating mixed sources. However, unlike the scenario faced by VAE which is unsupervised and source-agnostic, our task is not completely "blind", i.e., prior knowledge about a specifically interested component $\mathbf{y}$ is available. In this case, the variational lower bound should be $\mathbb{E}_{\tilde{q}(\mathbf{z}|\mathbf{y})}\left[\log\frac{p(\mathbf{y},\mathbf{z})}{\tilde{q}(\mathbf{z}|\mathbf{y})}\right]$, where $\tilde{q}(\mathbf{z}|\mathbf{y})$ is

an approximate posterior distribution in the form $\tilde{q}(\mathbf{z}|\mathbf{y}) = \int p(\mathbf{z}|\mathbf{x}) q(\mathbf{x}|\mathbf{y}) d\mathbf{x}$. It can be observed that the posterior $\tilde{q}(\mathbf{z}|\mathbf{y})$ takes into account the effect of the mixing process $q(\mathbf{x}|\mathbf{y})$, thus covering a broader class of distributions and becoming more expressive. We also notice that there is still the desired component $\mathbf{y}$ in the denominator. Considering the goal of extraction signal component from a nonlinear mixture, we would like to maximize $\mathbb{E}_{\tilde{q}(\mathbf{z}|\mathbf{y})}\left[\log \frac{p(\mathbf{y},\mathbf{z})}{q(\mathbf{z}|\mathbf{x})}\right]$ instead, in which case $\mathbf{x}$ and $\mathbf{y}$ should be used as the input and output of the network, respectively. Therefore, we need to study this new lower bound specifically.

**Lemma 3.1.** *We get a tighter lower bound for the source extraction problem:*

$$\log p(\mathbf{y}) \geq \mathbb{E}_{\tilde{q}(\mathbf{z}|\mathbf{y})}\left[\log \frac{p(\mathbf{y},\mathbf{z})}{q(\mathbf{z}|\mathbf{x})}\right] \geq \mathbb{E}_{\tilde{q}(\mathbf{z}|\mathbf{y})}\left[\log \frac{p(\mathbf{y},\mathbf{z})}{\tilde{q}(\mathbf{z}|\mathbf{y})}\right]. \tag{3}$$

*Proof.* For the left inequality, by using Jensen's inequality, we have:

$$\mathbb{E}_{\tilde{q}(\mathbf{z}|\mathbf{y})}\left[\log \frac{p(\mathbf{y},\mathbf{z})}{q(\mathbf{z}|\mathbf{x})}\right] = \int \tilde{q}(\mathbf{z}|\mathbf{y}) \log \frac{p(\mathbf{y},\mathbf{z})}{q(\mathbf{z}|\mathbf{x})} d\mathbf{z} = \mathbb{E}_{q(\mathbf{x}|\mathbf{y})} \mathbb{E}_{q(\mathbf{z}|\mathbf{x})}\left[\log \frac{p(\mathbf{y},\mathbf{z})}{q(\mathbf{z}|\mathbf{x})}\right]$$

$$\leq \log \mathbb{E}_{q(\mathbf{x}|\mathbf{y})} \mathbb{E}_{q(\mathbf{z}|\mathbf{x})}\left[\frac{p(\mathbf{y},\mathbf{z})}{q(\mathbf{z}|\mathbf{x})}\right] = \log p(\mathbf{y}). \tag{4}$$

For the right inequality, by using Jensen's inequality, we have:

$$\mathbb{E}_{\tilde{q}(\mathbf{z}|\mathbf{y})}\left[\log \frac{p(\mathbf{y},\mathbf{z})}{q(\mathbf{z}|\mathbf{x})}\right] = \mathbb{E}_{\tilde{q}(\mathbf{z}|\mathbf{y})}[\log p(\mathbf{y},\mathbf{z})] - \mathbb{E}_{\tilde{q}(\mathbf{z}|\mathbf{y})}\left[\log q(\mathbf{z}|\mathbf{x})\right]$$

$$\geq \mathbb{E}_{\tilde{q}(\mathbf{z}|\mathbf{y})}[\log p(\mathbf{y},\mathbf{z})] - \mathbb{E}_{\tilde{q}(\mathbf{z}|\mathbf{y})}[\tilde{q}(\mathbf{z}|\mathbf{y})] = \mathbb{E}_{\tilde{q}(\mathbf{z}|\mathbf{y})}\left[\log \frac{p(\mathbf{y},\mathbf{z})}{\tilde{q}(\mathbf{z}|\mathbf{y})}\right]. \tag{5}$$

$\square$

As a result, we can optimize the surrogate lower bound $\mathbb{E}_{\tilde{q}(\mathbf{z}|\mathbf{y})}\left[\log \frac{p(\mathbf{y},\mathbf{z})}{q(\mathbf{z}|\mathbf{x})}\right]$ of sVCD, which is tighter than that of VAE, for the task of source extraction. This variational inference can be readily implemented by the sVCD network in Section 4.

## 4  SUPERVISED VARIATIONAL COMPONENT DECODER ARCHITECTURE

To implement the variational inference framework for extracting signals from nonlinear mixture as presented in Section 3, we specifically design sVCD network to take observed nonlinear signal mixtures (containing multiple components) as input and to output the temporal sequence of the desired component. It inherits from encoder-decoder architecture with an encoder and decoder, both based on Recurrent Neural Network (RNN) (Cho et al., 2014), sandwiching a latent space. The architecture of sVCD is shown in Figure 2, and we elaborate the details in the following.

### 4.1  ENCODER

We employ an RNN-based encoder to implement the $q(\mathbf{z}|\mathbf{x})$ of the variational inference as introduced in Section 3. The encoder is parameterized by $\phi$. As demonstrated in Figure 3a, the RNN-based encoder $q_\phi(\mathbf{z}_{1:T}|\boldsymbol{x}_{1:T})$ of sVCD takes in the temporal signal $\boldsymbol{x}_{1:T} = [\boldsymbol{x}_1, \cdots, \boldsymbol{x}_t, \cdots, \boldsymbol{x}_T]$ of length $T$, where $\boldsymbol{x}_t$ is abbreviated for $\boldsymbol{x}_t \in \mathbb{R}^N$, $N$ represents number of observed signal mixtures. The encoder takes the responsibility of decomposing and selecting the desired component, which is in turn encoded to a compressed latent distribution $\boldsymbol{z}$. Samples drawn from $\boldsymbol{z}$ are then used to drive the decoder for recovering the desired component. We use RNN to form the encoder as it is naturally suitable for sequence modeling. At each timestamp $t$, the RNN reads the symbol $\boldsymbol{x}_t$ and previous hidden state denoted as $\boldsymbol{e}_{t-1}$ that summarizes the past information, and updates the new hidden state $\boldsymbol{e}_t$. In general, RNNs model sequences by parameterizing a factorization of the joint sequence probability distribution as a product of conditional probabilities such that:

$$p(\mathbf{x}_1, \mathbf{x}_2, \ldots, \mathbf{x}_T) = \prod_{t=1}^{T} p(\mathbf{x}_t|\mathbf{x}_{<t}), \tag{6}$$

$$p(\mathbf{x}_t|\mathbf{x}_{<t}) = g_\tau(\mathbf{e}_{t-1}), \tag{7}$$

where $g_\tau$ is a function that maps the RNN hidden state $\mathbf{e}_{t-1}$ to a probability distribution over possible outputs, and $\tau$ is the parameter set of $g$. RNN can be implemented with gated activation functions for longer dependecies, such as Long Short-Term Memory (LSTM) (Hochreiter & Schmidhuber, 1997) or Gated Recurrent Unit (GRU) (Cho et al., 2014). In this paper, we apply bidrectional GRU to for sequential learning. Given an input sequence $\boldsymbol{x}_{1:T}$ to the bidirectional GRU, we obtain the encoded features $\boldsymbol{e}_{1:T} = [\overrightarrow{\boldsymbol{e}_1}, \overleftarrow{\boldsymbol{e}_T}; \cdots ; \overrightarrow{\boldsymbol{e}_T}, \overleftarrow{\boldsymbol{e}_1}]$.

We let the prior distribution $\boldsymbol{z}_t \sim \mathcal{N}(\boldsymbol{\mu}_t, \boldsymbol{\Sigma}_t^2)$ be a Gaussian with a diagonal covariance matrix, whose mean and log-variance are parameterized by neural networks (usually implemented by fully connected layer), and the values depend on deterministic encoded features $\boldsymbol{e}_t$:

$$\boldsymbol{\mu}_t = \mathbf{W}^\mu \boldsymbol{e}_t + \mathbf{b}^\mu, \log \boldsymbol{\Sigma}_t^2 = \mathbf{W}^\Sigma \boldsymbol{e}_t + \mathbf{b}^\Sigma. \qquad (8)$$

Then $\boldsymbol{z_t}$ is sampled from mean and variance by reparameterization tricks (Kingma & Welling, 2013). We denote the encoder parameters for computing the input features as $\phi_x$ and the parameters for computing the mean and variance as $\phi_e$, we can model encoder as:

$$q_\phi(\boldsymbol{z}_{1:T}|\boldsymbol{x}_{1:T}) = \prod_{t=1}^{T} q_{\phi_e}(\boldsymbol{e}_t|\boldsymbol{x}_{<t}) \prod_{t=1}^{T} q_{\phi_z}(\boldsymbol{z}_t|\boldsymbol{e}_t). \qquad (9)$$

As such, the encoder has transformed the input temporal sequence to distributions on the latent space, which can be then readily sampled to drive the decoder for recovering the desired component $\boldsymbol{y}$.

## 4.2 ATTENTION MECHANISM

Although RNN introduced in Section 4.1 is a good way to handle sequential data, they may still lack the ability to discover very long dependencies across thousands of steps. To solve this problem, we employ the scaled dot product attention mechanism (Vaswani et al., 2017) to add more contexts to the temporal sequence; the modified encoder architecture is shown in Fig. 3b. The core idea of attention is to focus on the most relevant parts of the input sequence for each output. An attention function can be described as mapping a query and a set of key-value pairs to an output. Specifically, the query $\boldsymbol{q}_{1:T}$, key $\boldsymbol{k}_{1:T}$, and value $\boldsymbol{v}_{1:T}$ can be deemed as modified versions of the input $\boldsymbol{x}_{1:T}$ by three different linear transformations as follows:

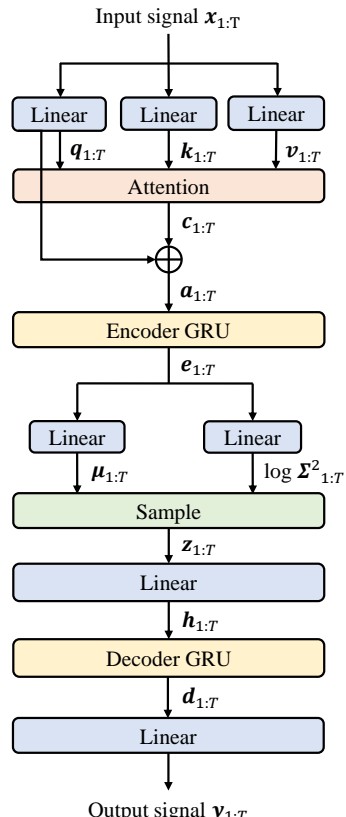

Figure 2: Architecture of sVCD.

$$\boldsymbol{q}_{1:T} = \mathbf{W}^q \boldsymbol{x}_{1:T} + \mathbf{b}^q, \qquad (10)$$

$$\boldsymbol{k}_{1:T} = \mathbf{W}^k \boldsymbol{x}_{1:T} + \mathbf{b}^k, \qquad (11)$$

$$\boldsymbol{v}_{1:T} = \mathbf{W}^v \boldsymbol{x}_{1:T} + \mathbf{b}^v, \qquad (12)$$

where $\mathbf{W}$ and $\mathbf{b}$ (with respective superscripts) are parameters that are trained to transform the input to its corresponding query $\boldsymbol{q}_{1:T}$, key $\boldsymbol{k}_{1:T}$, and value $\boldsymbol{v}_{1:T}$, whose dimensions are denoted by $d_q$, $d_k$, $d_v$. The output context $\boldsymbol{c}_{1:T}$ is obtained by calculating a weighted sum of the values, where the weight of each value is a normalized product of the query and its corresponding key. Then the context is concatenated to the query to form the final output of our attention layer. In summary, the attention mechanism can be expressed as follows:

$$\boldsymbol{c}_{1:T} = \text{softmax}\left(\frac{1}{\sqrt{d_k}}\boldsymbol{q}_{1:T}\boldsymbol{k}_{1:T}^T\right)\boldsymbol{v}_{1:T}, \qquad (13)$$

$$\boldsymbol{a}_{1:T} = (\boldsymbol{c}_{1:T} \oplus \boldsymbol{q}_{1:T}), \qquad (14)$$

where $\oplus$ represents concatenation. Compared with its input $\boldsymbol{x}_{1:T}$, the output of the attention layer $\boldsymbol{c}_{1:T}$ contains more contexts of the whole signal by taking into account the interactions among the elements, thus forming a better input for the ensuing RNN.

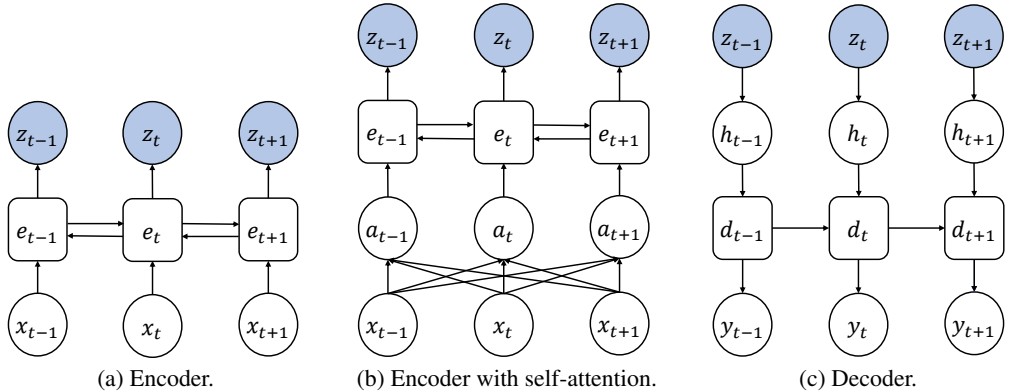

(a) Encoder.  (b) Encoder with self-attention.  (c) Decoder.

Figure 3: Architecture of the encoder and decoder.

## 4.3 DECODER

To realize $p_\theta(\boldsymbol{y}|\boldsymbol{z})$ in the variational inference problem discussed in Section 3, we implement a decoder using RNN-based network, as shown in Figure 3c. The decoder $p_\theta(\boldsymbol{y}_{1:T}|\boldsymbol{z}_{1:T})$ is parameterized by $\theta$, and serves to reconstruct the source of interest $\boldsymbol{y}_{1:T}$ from a latent vector $\boldsymbol{z}_{1:T}$. The decoding consists of three steps. Firstly the latent vector $\boldsymbol{z}_{1:T}$ is expanded to a hidden feature dimension by a fully connected layer:

$$\boldsymbol{h}_t = \mathbf{W}^h \boldsymbol{z}_t + \mathbf{b}^h, \tag{15}$$

with $\mathbf{W}^h$ and $\mathbf{b}^h$ being the weight matrix and bias represented by the linear layer. Then $\boldsymbol{h}$ is sent to the sequence decoding layer implemented by a forward directional GRU that produces output $\boldsymbol{d}$. Finally, we have the decoded output sequence $\boldsymbol{d}_{1:T} = [\overrightarrow{\boldsymbol{d}_1}, \cdots, \overrightarrow{\boldsymbol{d}_t}, \cdots, \overrightarrow{\boldsymbol{d}_T}]$, where feature at each step is connected to a two linear layers with rectified linear units (ReLU) to the dimension of $\boldsymbol{y}_{1:T}$.

$$\boldsymbol{y}_t = \mathbf{W}^{y2}(\mathrm{ReLU}(\mathbf{W}^{y1}\boldsymbol{d}_t + \mathbf{b}^{y1})) + \mathbf{b}^{y2}, \tag{16}$$

where $\mathbf{W}^{y1}$, $\mathbf{W}^{y2}$, $\mathbf{b}^{y1}$, and $\mathbf{b}^{y2}$ are the weights and biases of the linear layer. Overall we have the decoder represented as:

$$p_\theta(\boldsymbol{y}_{1:T}|\boldsymbol{z}_{1:T}) = \prod_{t=1}^{T} p_{\theta_h}(\boldsymbol{h}_t|\boldsymbol{z}_t) \prod_{t=1}^{T} p_{\theta_d}(\boldsymbol{d}_t|\boldsymbol{h}_{<t}) \prod_{t=1}^{T} p_{\theta_y}(\boldsymbol{y}_t|\boldsymbol{d}_t). \tag{17}$$

## 4.4 LOSS FUNCTIONS

As stated in Section 3, we maximize the variational lower bound $\mathbb{E}_{\tilde{q}(\mathbf{z}|\mathbf{y})}\left[\log \frac{p(\mathbf{y},\mathbf{z})}{q(\mathbf{z}|\mathbf{x})}\right]$ in Equation (3) to solve the variational inference problem. Since we have parameterized the probabilities with deep neural networks, we rewrite the variational lower bound as $\mathbb{E}_{\tilde{q}(\mathbf{z}|\mathbf{y})}\left[\ln \frac{p_\theta(\mathbf{y},\mathbf{z})}{q_\phi(\mathbf{z}|\mathbf{x})}\right]$, and it can be decomposed as:

$$\mathbb{E}_{\tilde{q}(\mathbf{z}|\mathbf{y})}\left[\ln \frac{p_\theta(\mathbf{y},\mathbf{z})}{q_\phi(\mathbf{z}|\mathbf{x})}\right] = \mathbb{E}_{q_\phi(\mathbf{x}|\mathbf{y})}\mathbb{E}_{q_\phi(\mathbf{z}|\mathbf{x})}\left[\ln \frac{p_\theta(\mathbf{y},\mathbf{z})}{q_\phi(\mathbf{z}|\mathbf{x})}\right]$$

$$= \mathbb{E}_{q_\phi(\mathbf{x}|\mathbf{y})}\left[\mathbb{E}_{q_\phi(\mathbf{z}|\mathbf{x})}[\ln p_\theta(\mathbf{y}|\mathbf{z})] - \mathbb{KL}\left(q_\phi(\mathbf{z}|\mathbf{x})|p_\theta(\mathbf{z})\right)\right]. \tag{18}$$

This lower bound can be separated into two parts, the first term is a reconstruction error between the sVCD output and the ground truth, and the second term as a regularizer which enforces the approximate posterior to be close to the specified prior. To practically implement the reconstruction loss $\mathbb{E}_{q_\phi(\mathbf{z}|\mathbf{x})}[\ln p_\theta(\mathbf{y}|\mathbf{z})]$, we use the $L_2$ distance between the separated component $\hat{\mathbf{y}}$ and $\mathbf{y}$:

$$\mathcal{L}_{\mathrm{RC}} = \|\hat{\mathbf{y}} - \mathbf{y}\|^2. \tag{19}$$

Since the prior distribution is defined to be unit Gaussian, the second term can be practically implemented by:

$$\mathcal{L}_{\mathrm{RE}} = \mathbb{KL}\left(\mathcal{N}\left(\boldsymbol{\mu}_{\mathrm{t}}, \boldsymbol{\Sigma}_{\mathrm{t}}^2\right), \mathcal{N}(\mathbf{0}, \mathbf{I})\right). \tag{20}$$

And the loss function of sVCD can be defined as:

$$\mathcal{L}_{\mathrm{sVCD}} = \mathcal{L}_{\mathrm{RC}} + \lambda \mathcal{L}_{\mathrm{RE}}, \tag{21}$$

where $\lambda$ is the weight controlling the strength of the regularization.

## 5 EVALUATION

In this section, we first report experimental results on three datasets for sVCD compared with state-of-the-art baselines. We then perform an ablation study on sVCD by removing its certain parts, in order to gaining more understanding of the network.

### 5.1 DATASETS FOR NONLINEAR MIXTURES

Traditional datasets for BSS, e.g., those created for audio source separation (Garofolo et al., 1993; Vincent et al., 2007) are mostly linear. To demonstrate the nonlinear mixture separation efficacy of sVCD, we employ three datasets of nonlinear mixtures, with details elaborated in the following.

**Artificially Generated Dataset** One artificially generated dataset is employed in our evaluation due to its special statistical properties. For this dataset, both source signals and observed signals are generated according to the settings in (Hyvärinen & Morioka, 2017). All source signals are randomly produced from an AutoRegressive (AR) model with a non-Gaussian process. Then, we obtain nonlinear mixtures from the source signals leveraging a nonlinear mixing function comprised of a multi-layer perceptron (MLP) with nonlinear activation functions, i.e., leaky $\mathrm{ReLU}$ units. For each of the 4,000 trials, we leverage the model to generate 12 temporally dependent sources (independent of each other) that are in turn mixed to produce 12 observed signals; we select one as source of interest to be extracted by sVCD.

**RF-PPG Dataset** Contact-free vital sign sensing is the problem of extracting vital signs of human subjects via contact-free sensors, a representative and practical scenario for signal extraction from the nonlinear mixture. To be specific, the micromovements of the skin caused by heartbeats are mixed non-linearly with respiration-induced movements and other body movements, and the mixture is carried on RF signals reflected back from human bodies. In this dataset, a high-resolution radar sensor (Novelda AS, 2017) is used to collect the RF signals. The ground truth heartbeat photoplethysmography (PPG) waveforms are measured by NeuLog (2017). The data collection is conducted with 12 healthy subjects (6 females and 6 males) under the IRB approval of our institute. The subjects are asked to sit at quasi-static and natural states. The RF-PPG dataset includes 48-hour data, approximately 180,000 heart cycles in total, and is divided into 20-seconds segments.

**EEG-EOG Dataset** Electroencephalography (EEG) is an electrophysiological monitoring technique for recording electrical activity on the scalp, which has been proven to reflect the activity of the brain's surface layer underneath (Henry, 2006). Due to the sophisticated structure of the brain, the signals at different locations of the underneath brain (whose ground truths cannot be obtained directly), when manifested as EEG signals, are often superimposed nonlinearly and exhibit complex behavior (Steyrl et al., 2013), and the nonlinearity is escalated by the nonlinear properties of the brain (Nunez et al., 2006) and the skull (Zhang et al., 2014). Moreover, the electrical activity of the eye, which is the main source of artifacts of brain signals, complicates the EEG recordings. As such, it is often required to isolate and remove electrical signals of the eyes, i.e., electrooculography (EOG) signals, from EEG recordings. In the collected dataset (Torkamani-Azar et al., 2020), EEG and EOG signals are recorded simultaneously to form 2.5-minute segments. The EEG signals have 64 channels corresponding to electrodes mounted on the scalp, and the EOG signals have 3 channels corresponding to electrodes connected to the outer corners of both left and right eyes, and the middle of the eyebrows. The dataset is collected from 10 human subjects for a duration of 105 minutes.

## 5.2 EXPERIMENT SETUP

We employ the aforementioned datasets to evaluate the performance of sVCD. The collected datasets are divided into 80% training and 20% test sets. For the model parameters, we set the size for $d_q$, $d_k$ and $d_v$ as 64; $e$, $z$, $h$, $d$ are set respectively as: 64, 8, 128, 128; and dropout rate at 0.2. The weight $\lambda$ is set to 0.1 by line search. For the training process, we employ the Adam optimizer (Kingma & Ba, 2014) with a starting learning rate of 0.001 to update the parameters according to equation 21. The batch size is set to 20. To enable proper evaluation, all signals are scaled to $[-1, 1]$ beforehand. To evaluate the performance of the method we calculate the cosine similarity between the recovered component $\hat{\mathbf{y}}$ and the ground truth component $\mathbf{y}$. The cosine similarity is defined as $\mathcal{S}(\hat{\mathbf{y}}, \mathbf{y}) = \frac{\hat{\mathbf{y}} \cdot \mathbf{y}}{\|\hat{\mathbf{y}}\|\|\mathbf{y}\|} = \frac{\sum_{t=1}^{T} \hat{y}_t y_t}{\sqrt{\sum_{t=1}^{T} \hat{y}_t^2} \sqrt{\sum_{t=1}^{T} y_t^2}}$, with value in the range of $[-1, 1]$. A high similarity indicates that better performance of the model, and vice versa.

To compare sVCD with previous works, we choose ICA (Hyvärinen & Oja, 2000) and Conv-TasNet (Luo & Mesgarani, 2019) as baseline methods; they are both designed for separating linear signal mixture. ICA is chosen because it is a widely used blind source separation technique, and the fitting of ICA and the training of sVCD are similar since both consider the context of the signal including both the past and the future. Conv-TasNet is chosen as the baseline because it is a state-of-the-art deep-learning method for automatic signal separation. We follow the parameter settings in the original papers, and for ICA, we separate 12, 5, and 64 independent components from the artificially generated, RF-PPG, and EEG-EOG datasets, respectively.

## 5.3 RESULTS

In Figure 4, we show the waveforms of the ground truth, and the signals extracted by ICA, Conv-TasNet (abbreviated as C-TasNet) and sVCD signal on the three datasets. For the artificially generated data, sVCD retrieves a component approximately the same as the ground truth, while ICA and Conv-TasNet obtain very noisy results. For the RF-PPG data, sVCD successfully recovers the heartbeat waveform bearing correct heart rate, while ICA barely separate heartbeat signal from respiration as the breath waveform is still dominating, and Conv-TasNet fails to recover event details in the waveform. For the EEG-EOG dataset, sVCD extracts a source signal bearing high similarity to the ground truth, while ICA and Conv-TasNet seem to have deduced totally wrong outcomes.

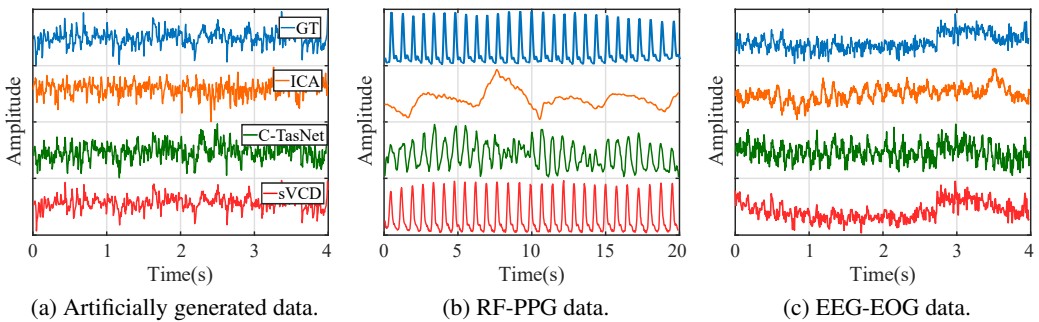

Figure 4: Source signals extracted by ICA, Conv-TasNet, and sVCD, against the ground truth (GT).

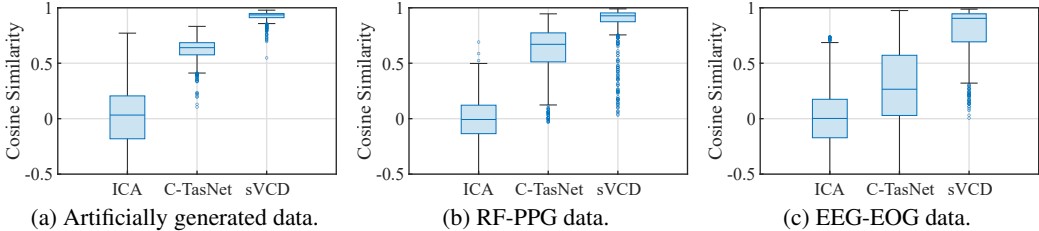

Figure 5: Statistics on cosine similarities for respective datasets.

In Figure 5, we show the cosine similarities between the recovered waveforms and the ground truth of ICA, Conv-TasNet, and sVCD. It can be observed that the performance of sVCD substantially surpasses those of ICA and Conv-TasNet on all three datasets, and the median cosine similarities of sVCD are all above 0.8 on all three datasets, indicating a strong resemblance between the sVCD extracted source signals and the ground truth.

## 5.4 ABLATION STUDIES

We provide an ablation study of the main design decisions for sVCD in this Section. In the first study labeled as "Seq2Seq", we skip the variational inference part, and make the output of the encoder GRU directly driving the decoder GRU. Meanwhile, in the second study "sVCD with only mean regularization", instead of regularizing the distributions to the standard Gaussian, we regularize only the mean value towards 0. Lastly, the study labeled as "sVCD without attention" holds out the attention mechanism. Our goal is to see the respective accuracy drops to measure their importance in the proposed sVCD network. Given the cost of training a single model, we did not compute confidence intervals for each variation. The result of the ablation study is shown in Table 1.

We can observe that our proposed sVCD achieves cosine similarities of 0.9358, 0.8634, and 0.7803 on the three datasets, respectively. Performance drops can be observed for the Seq2Seq model on Artificially generated, RF-PPG and EEG-EOG datasets, indicating that a discrete latent space without regularization encodes less useful information that can be used by the decoder to recover the source. For the sVCD with only mean regularization, further performance decrease can be observed, proving that the distribution mean and variance should better off be regularized together. As for the sVCD without attention, some clear drop in cosine similarity can be observed on all three datasets, demonstrating that the attention mechanism helps boosting the performance of sVCD.

Table 1: Average cosine similarities in the ablation studies.

|  | Artificially generated | RF-PPG | EEG-EOG |
|---|---|---|---|
| Proposed sVCD | 0.9358 | 0.8634 | 0.7803 |
| Seq2Seq | 0.9339 | 0.8593 | 0.7765 |
| sVCD with only mean regularization | 0.8737 | 0.8499 | 0.7596 |
| sVCD without attention | 0.8552 | 0.8342 | 0.7577 |

## 6 CONCLUSION

Taking an important step toward deep learning enabled signal processing, we proposed a new approach for source extraction from nonlinear mixture. We started by formulating nonlinear signal extraction as a modified variational inference problem. By detailed derivation, we have proved that the mathematical framework of signal extraction can be implemented by an sVCD network, in which the modified variational lower bound is maximized using backpropagation. Experiments on artificially generated data, RF-PPG data, and EEG-EOG data have proved the signal extraction efficacy of sVCD. The sVCD in this paper is used as a vehicle to validate the overall mathematical framework, but better architectures are likely to be found. In the future, we plan to apply the sVCD network to other promising applications such as source extraction from video data, econometric data, and biomedical data (such as EMG and ECG), in which source signals have nonlinear mixtures.

### ETHICS STATEMENT

All experiments involve human subjects have been approved by IRB of our institute, and the experiments have been conducted strictly followed the standard procedures. The RF-PPG data will not be made available to the public to protect the privacy of the subjects.

### REPRODUCIBILITY STATEMENT

In this section we briefly reference the details needed to help reproducibility. For the dataset: we have uploaded the artificially generated dataset in the link in our supplementary materials; due to

privacy concerns we could not make the RF-PPG dataset to public; the EEG-EOG dataset is a public dataset acquired from Torkamani-Azar et al. (2020). The model implementation, settings are described as in Section 5.2. We have also uploaded the source code to allow for reproducing our results in supplementary materials.

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

# A   EXPERIMENTS ON LENGTH OF SEQUENCE

As sVCD is built on RNN structure, there is no strict requirement on the length of input sequence. We provide an experiment on the adaptability of sVCD by training with sequence length of $L$, then testing with different lengths from $2L$ to $0.01L$. For the three datasets mentioned in Section 5.1, $L$ is 800 time steps which equivalent to: 16 seconds for artificially generated data, 20 seconds for RF-PPG data, 4 seconds for EEG-EOG data. The cosine similarities are presented in Figure 6. It can be observed that generally tested with sequence lengths from $0.5L$ to $2L$, resulting statistics do not have much variation for all three datasets. For artificially generated dataset, test data length with only $0.01L$ still has median of similarity above 0.9, but much larger error can be observed. For RF-PPG dataset, test data that have length lower than $0.4L$ have similarities much lower than those longer. While for EEG-EOG dataset, test data length lower than $0.5L$ will make the performance drops. Different minimum sequence length of test data may be due to different source signals have distinct time-correlation features. Therefore, it is better to take a sufficiently long sequence as it is hard to judge the minimum yet sufficient length a priori. Overall, the results show a promising future of sVCD that after training with an adequate length, the model could adapt to not only short time observation but also long time monitoring without extra training efforts.

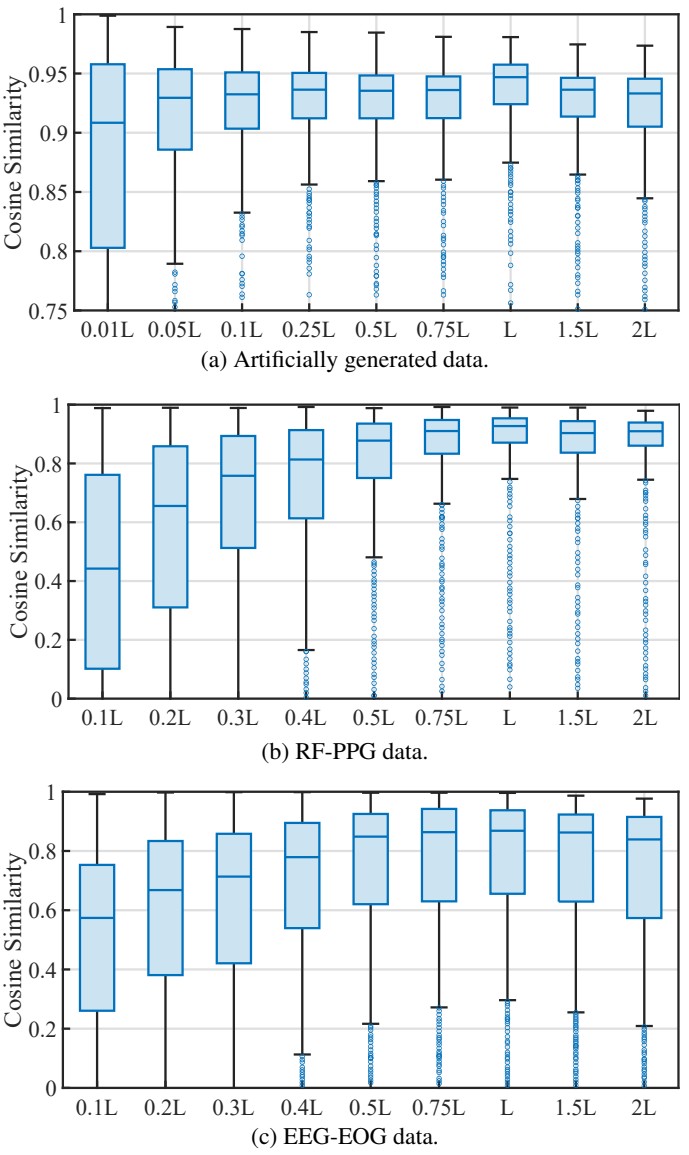

Figure 6: Statistics on cosine similarities for respective testing data sequence length.

# B  EXPERIMENTS ON LINEAR AND NONLINEAR REGRESSIONS

We have performed linear and nonlinear regression for the signal extraction purpose. For nonlinear regression, we employ least square regression and support vector regression. For each category of nonlinear regression, we employ polynomial, radial basis function and sigmoid kernel. For nonlinear regression, we only report the results of support vector regression with polynomial kernel because this combination delivers the best performance. The recovered waveforms of linear regression (denoted as "LinReg"), nonlinear regression (denoted as "NLinReg"), and sVCD are shown in Figure 7, and their respective cosine similarities are shown in Figure 8. It can be observed that sVCD significantly outperforms these two regression methods, while both regressions fail to recover meaningful results. The reason for their inferior performance can be readily explained by their omission of temporal correlations, but they do offer instantaneous point-to-point conversion instead of the seq-to-seq manner operated by sVCD.

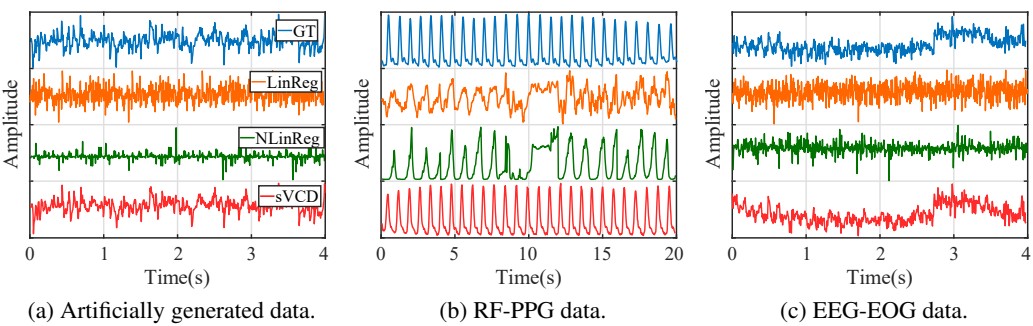

(a) Artificially generated data.    (b) RF-PPG data.    (c) EEG-EOG data.

Figure 7: Source signals extracted by linear regression, nonlinear regression, and sVCD, against the ground truth (GT).

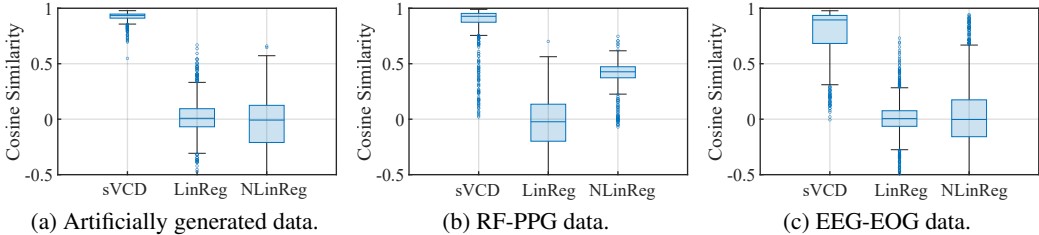

(a) Artificially generated data.    (b) RF-PPG data.    (c) EEG-EOG data.

Figure 8: Statistics on cosine similarities for respective datasets.

