# OpenReview forum: "Variational Component Decoder for Source Extraction from Nonlinear Mixture"
_ICLR.cc/2022/Conference — ICLR 2022 Submitted_

### Official Review · Reviewer_aZ4J · 2021-10-30

**Correctness:** 3
**Technical Novelty And Significance:** 3
**Empirical Novelty And Significance:** 2
**Recommendation:** 3
**Confidence:** 4

**Main Review:**

The paper is well written and clear to understand. Proposed method is an elegant extension of a seq-to-seq VAE that exploits an attention mechanism, and reconstructs a source signal (that one needs to have available as a ground truth in the training set -which is a limitation-), based on a variational lower bound on the source signal’s data likelihood. Ablation studies are interesting and nicely performed, as they demonstrate a significant benefit of using an attention mechanism for time-series signal source extraction. In my opinion the paper’s main weakness is in the experimental analyses and evaluations that are presented.

Authors tried to make a distinction between nonlinear “source separation” and “source extraction” from mixtures. I am not fully convinced on the arguments around this theme. (1) For instance authors present arguments on “significantly reducing the computation cost”, when one is interested only in extracting the source signal of interest (i.e., nonlinear source extraction), rather than performing nonlinear blind source separation. This should be supported later in the experiments (which the authors did not), by discussing computation costs of the proposed algorithm, with respect to e.g., ICA. (2) Another line of argument by the authors state that “existing work on representation disentanglement focuses on image or video data, where signal mixtures never take place”. I do not fully agree on this statement, as in most cases these architectures are also applicable to time-series signal recordings (cf., studies on variational speech separation). To support their claim, authors can maybe present a baseline comparison to a simple VAE-based source separation model.

Performed experiments on the EEG-EOG dataset do not make much sense from a practical viewpoint, and reveal an important weakness. From what the reader can understand, the ground truth “y” to train the sVCD models for reconstruction is given as the EOG signals that are provided in the dataset. This means that sVCD recovers the best estimate of the EOG time-series from nonlinear EEG mixtures. While it is successfully shown that sVCD can perform this, it is not a meaningful experiment to demonstrate. In nonlinear EEG source separation/extraction, one is always interested in recovering some EEG source signal (e.g., originating from an independent cortical source component), that is clean from artifactual (e.g., EOG, EMG) mixing. I understand that since there is no given ground truth of any EEG source component, it would also be not possible to demonstrate such an experiment with sVCD. However, I do not understand why recovering EOG artifacts from EEG signals would be an interesting demonstration to include in the paper.

Overall, methodological comparisons seem shallow. It is also not clear how did the authors implement ICA (as a comparison baseline) in their experiments? How many independent components were initially extracted and how were the ICs selected following the decomposition? No supplementary or further descriptions of these experiments are presented. For instance, for EEG-EOG experiments, if one decomposes multi-channel EEG into independent components by pooling the training data, it should be generally obvious to recover the EOG component in one of the ICs besides cortical EEG signal source estimates (this is a widely-used approach for EOG artifact removal to discard ICs relevant to eye-blinks [T.-P. Jung et al., NIPS 1997], [S. Makeig et al., PNAS 1997]).

Minor comments:
- How did the authors determine the KL-term regularization strength? It is also not mentioned what are the parameters used?
- “sour” -> “source” on page 2, fourth line.
- “realized” -> “realize” on page 6, first sentence.


**Summary Of The Paper:**

Authors propose a supervised variational component decoder (sVCD) framework that estimates a single source signal sequence from non-linear mixtures. Proposed model relies on a sequence-to-sequence translating variational encoder-decoder architecture, where optimization is performed based on a variational lower bound on the source signal’s data likelihood. Experiments are performed on artificially generated nonlinear sequence mixtures, RF sensing data and an EEG-EOG dataset.

**Summary Of The Review:**

The paper is clearly written and organized. However the paper is rather weak in depicting its contributions with the present experimental analyses and evaluations, and lacks detailed comparisons to other methods. I listed my major concerns in the main review, and would be willing to re-address my rating based on the authors’ responses and revisions.

---

> ### Author Response · Authors · 2021-11-15
> **Response to Reviewer aZ4J**
>
> We would like to thank the reviewer for constructive feedbacks and valuable comments. We address your comments below.
>
> We are grateful to the reviewer for pointing out our misleading statement. In fact, as no existing blind source separation method works for nonlinear mixture, our statement is more a conjecture that, if there existed a blind source separation algorithm based on deep learning, its computational cost should be greater than sVCD that only extracts one component out of a nonlinear mixture. We have revised the introduction of our paper by making this point clear. Note that it is not appropriate to compare the complexity of sVCD with ICA, as the latter cannot handle nonlinear mixture and is not even a deep learning approach.
>
> We have revised our texts to clarify that the methods for disentangling for image and video data do not apply since they focus on disentangling on the representation, rather than disentangling a signal mixture. Moreover, methods for speech disentanglement are also not applicable for nonlinear mixtures, since these signals are superposed in a linear fashion [R1]. In order to prove this statement, we have added a comparison baseline in Section 5.3.
>
> [R1] Luo, Yi, and Nima Mesgarani. "Conv-TasNet: Surpassing Ideal Time–frequency Magnitude Masking for Speech Separation." IEEE/ACM Transactions on Audio, Speech, and Language Processing 27, no. 8 (2019): 1256-1266.
>
> We would like to clarify that recovering EOG from EEG is meaningful: by extracting EOG from EEG signals, we can better understand their relationship, and the extracted EOG signal can possibly help remove interference caused by eye activities in EEG.
>
> We have clarified in the revision that we follow the standard procedures of ICA as stated in [R2]. In addition, 12, 5, and 64 independent components are separated from the artificially generated, RF-PPG, and EEG-EOG datasets, respectively. After performing ICA, we pick the independent components that resemble ground truth the most for evaluation. We also thank the reviewer for pointing out two relevant references [R3] and [R4]. It is true that the authors in those papers stress the influence of EOG in the ICA components. Unfortunately, their results (e.g., Fig. 2 in [R3] and Fig. 1 and 2 in [R4]) clearly demonstrate that ICA is unable to identify a component sharing a similar pattern with the EOG signal, which is actually proving the ineffectiveness of ICA in handling nonlinear mixture.
>
> [R2] Hyvärinen, Aapo, and Erkki Oja. "Independent Component Analysis: Algorithms and Applications." Neural Networks 13, no. 4-5 (2000): 411-430.
> [R3] Makeig, Scott, Tzyy-Ping Jung, Anthony J. Bell, Dara Ghahremani, and Terrence J. Sejnowski. "Blind Separation of Auditory Event-related Brain Responses into Independent Components." Proceedings of the National Academy of Sciences 94, no. 20 (1997): 10979-10984.
> [R4] Sejnowski, Terrence J. "Independent Component Analysis of Electroencephalographic Data." In Prof. of NeurIPS, vol. 8, p. 145. MIT press, 1996.
>
> Minor Points:
>
> We have revised our manuscript by correcting typos in text. The KL-term regularization strength was determined by line search, we provide the value used for evaluation in Section 5.2.

---

> > ### Comment · Reviewer_aZ4J · 2021-11-22
> > **Thanks for the responses**
> >
> > Thanks to the authors for their responses. I have carefully read the revisions, and I have decided to keep my rating as it is.
> >
> > It seemed like several clarifications in the MS were necessary, as performed with the revisions by the authors. I still think that the baseline linear ICA is not the only methodological comparison that the authors could have presented, while there exist more recent studies that explore methods for non-linear ICA problems such as: https://arxiv.org/pdf/1710.05050.pdf. I am also not convinced by the arguments around the EEG-EOG experiments. The authors state: “We can better understand their relationship, and the extracted EOG signal can possibly help remove interference caused by eye activities in EEG”. What is however done in reality is that one places a bio-potential sensing electrode (same as the EEG recording ones) under and/or above the eyes to record EOG, such that this signal can directly be used to extract EOG-free cortical sources of interest. Hence, I think being able to recover EOG from only-EEG recordings is not a reasonable challenge in my opinion.

---

> > > ### Author Response · Authors · 2021-11-23
> > > **Response to Reviewer aZ4J**
> > >
> > > We have made another round of revision, in which we have added two more baselines, namely linear and nonlinear regressions, for further comparison in the Appendix B. We hope to convince the reviewer that these comparison baselines, together with Conv-TasNet and ICA, are sufficient for evaluating sVCD. As for the nonlinear ICA mentioned by the reviewer, that GAN-based contrastive learning idea actually stems from Hyvarinen & Morioka, 2016&2017 cited in our original texts. According to our experience with them, the training cost is tremendous compared with sVCD: tens to hundreds of times longer in training time is one thing, while the difficulty in designing proper data augmentation schemes for constructing contrastive datasets is another. As a result, we only argue in the beginning of our paper that ICA-like full separation methods are far less efficient than our method when coming to extraction a few sources. After all, it might not be necessary for a proposal to compare against all methods in the literature; only relevant and comparable ones would be sufficient: our current baselines are either methodology-wise relevant (e.g., regression and Conv-TasNet) or complexity-wise comparable (e.g., linear ICA), of course thanks to the constructive suggestions from the reviewers.
> > >
> > > We fully agree that EOG can be measured directly in practice, but that would require extra instruments. Therefore, we still believe extracting EOG from EEG is a relevant contribution because it reveals the intrinsic relationship between the two. In fact, we believe that the discrepancy in judging the value of EOG extraction (even if it could persist after this round of response) shouldn’t devaluate the relevance of our contribution in sVCD. Therefore, we would sincerely hope the reviewer to kindly consider adjusting the rating score, taking our revision efforts and rebuttal arguments into account.

---

> > > > ### Comment · Reviewer_epBn · 2021-11-28
> > > > **if you don't measure it how will you train the model?**
> > > >
> > > > "We fully agree that EOG can be measured directly in practice, but that would require extra instruments. Therefore, we still believe extracting EOG from EEG is a relevant contribution because it reveals the intrinsic relationship between the two"
> > > >
> > > > This reasoning doesn't make sense in a typical EEG experiment. If the EOG weren't measured then we wouldn't be able to train the proposed Seq2Seq model to extract if from the EEG. Unless the paper is claiming it is able to perform subject independent EEG to EOG prediction, which would be significant.
> > > >
> > > > If it were blind source separation then yes, one could extract a component from EEG that is related to EOG, but that is not what is proposed.

---

> > > > > ### Author Response · Authors · 2021-11-30
> > > > > **Maybe a misunderstanding here**
> > > > >
> > > > > We believe there might a misunderstanding on the earlier interactions between Reviewer aZ4J and us. To our understanding, Reviewer aZ4J was questioning the necessity of extracting EOG out of the measured EEG-EOG mixture, as it is clear EOG can be directly measured. However, our argument is that, as measuring EOG requires extra instruments, our solution treats the additionally measured EOG as training data so that we may later directly extract EOG out of EEG-EOG mixture, without the need for extra instruments. In short, our argument is strictly in line with your point, i.e., using directly measured EOG as training data for sVCD.

---

### Official Review · Reviewer_epBn · 2021-11-01

**Correctness:** 3
**Technical Novelty And Significance:** 3
**Empirical Novelty And Significance:** 2
**Recommendation:** 5
**Confidence:** 4

**Main Review:**

Strengths:
The modeling problems the approach seeks to solve are interesting.

The papers review of nonlinear source separation and linear source separation helped set the context.

Approaches for Seq2Seq are likely to have an increasing impact in time-series tasks and demonstration of their potential is likely to have an impact in the machine learning for signal processing community.

The paper's architecture (combination of self-attention, recurrent network, generative code layer, and decoder) appears to be useful for sequence regression problems.

The experimental results look promising in their accuracy.

Weaknesses:
0. The problem boils down to a regression task. However, there are no baseline comparisons with nonlinear regression models. It is not clear how the ICA baseline, which could extract multiple linearly demixed source signals, is fairly compared. Is the ICA component shown in the figures and the best plots the best matched?  ICA is not only a linear approach but it also instantaneous. It doesn't consider past (or future values). Why couldn't MSE be used instead of cosine similarity? Is the model not able to match the mean or magnitude of the target?

1. Much of the time-series aspects are ignored.
a) The paper's architecture does consider nor impose the causal structure on the decoder. That is the self attention creates a non-causal representation.
b) The effect of the length of the sequences in the regression performance is not discussed. Are all training sequences the same length? Can they vary in testing?
c) Point about comparing against instantaneous ICA.

2. The EEG-EOG Dataset introduces an inaccurate description of electrophysiology. The electric potentials are instantaneously mixed. The reference cited for 'superimposed nonlinearly' actual refers to the non-linear dynamics not the combination of multiple sources impinging on the electrodes.  While nonlinearities can appear due to the sensing being nonlinear (amplifiers being saturated), this is not the sense of the description.

3. In the ablation study, it appears the Seq2Seq method performs well without the variation inference. However some gap exists and it isn't clear if this could be lowered by keeping the affine transform from the encoding and still regularizing the difference from mean 0.

Minor points:
page 1, "proposal mostly focus" -> "proposal mostly focus"
page 2, "sour extraction"
Figure 2, where are the time indices in the representation fo the encoder and prior distribution?
page 3, after equation 1 shouldn't the phrasing be that the KL divergence can be minimized by maximizing the variational lower bound?
page 5, the subscript notation of $1:T$ is not introduced and would seem to be a matrix. Generally the notation follows MATLAB/Octave but it is not clear to the reader.  For instance in equation 13, I am guessing the softmax operates on elements of the outer product by normalizing across each row.  The concatenation notation used in 14 should be defined.
page 5, first line "To be realized
page 5, "forward bi-directional" ?
page 5, $L^2$ distance is actually the squared $L_2$ distance between the signals (or $\ell_2$ in $\mathbb{R}^T$)

**Summary Of The Paper:**

The paper proposes an approach for supervised non-linear regression for multivariate time-series using a sequence-to-sequence approach with self-attention and generative prior on the latent codes. The paper poses this as a source extraction from a nonlinear mixture.

The paper shows how this can be applied to a synthetically created data set as well as two real world data sets.  The first is heartbeat and respiration from radio frequency.  The second is  EEG with  EOG and the goal is to remove the EOG from the EEG. The paper shows that the model is able to extract the signals in the example cases.

**Summary Of The Review:**

While the paper presents a meaningful architecture for the task, key aspects of the nature of time-series are ignored and no meaningful baselines are used for benchmarking. Thus the paper is preliminary and needs a more thorough set of comparisons against other non-linear regression problems with and without causal restrictions on the modeling.

-----------------------------

The authors have performed additional comparisons that help set the context. I am willing to raise my score if the text is further clarified to include the fact that is this is a non-linear multiple input and single output system identification model.

I agree the concerns with non-causality may not be valid in all settings, and the additional testing on performance at shorter time segments is important.

Finally, I would like to encourage the revision to have careful wording regrading the non-linearity of the mixing versus non-linearity of processes. These two can get conflated. I agree that sometimes electrical recordings of linearly mixed sources result in nonlinear mixtures due to non-stationaries due to sensor/electrode movement as well as amplifier/recording nonlinearities.

Further empricial analysis into the performance under challenging non-linearities such as clipping would be encouraged.

---

> ### Author Response · Authors · 2021-11-15
> **Response to Reviewer epBn**
>
> We would like to thank the reviewer for constructive feedbacks and valuable comments. We address your comments below.
>
> W0) We fully agree with the reviewer that our sVCD does perform a regression task, as the overall machine learning boils down to either regression or classification. Since there is no existing method for extraction from nonlinear mixture, ICA is chosen as the baseline because it is a widely used blind source separation technique. Moreover, ICA is indeed fairly compared. First, the ICA components evaluated and shown in figures are the best matched. Also, although ICA is runtime instantaneous, sVCD can also handle short segments of input signal (hence only causing very short delays). In addition, the fitting of ICA and the training of sVCD are similar since both consider the context of the signal including both the past and the future. We have revised our paper in Section 5.2, justifying our choice of ICA as the comparison baseline and adding more details of how we implement ICA in the experiment.
>
> MSE can certainly be used for sVCD because the loss function used for training sVCD is L_2 norm, which is equivalent to MSE. However, to the best of our knowledge, recovering the mean of a signal is not the strength of ICA, since this feature doesn’t concern statistical independence. Therefore, to enable a fair comparison, we choose the more representative cosine similarity for both sVCD and ICA in the evaluation.
>
> W1-a) We would like to argue that causality is not a necessary condition for our task. Although the time series is causal during its a priori generation (i.e., no information should flow from the future to the present), we don’t have to stick to the rule after obtaining the data, as posterior knowledge of the future can help us understand and extract the signal. For example, transformer [R1] and BERT [R2] both leverage non-causal structures for understanding and processing time series. Consequently, we employ the non-causal self-attention mechanism in the encoder. Moreover, in the ablation study, it can be observed that our proposed sVCD (equipped with the non-causal self-attention) outperforms Seq2Seq assuming pure causality. Nonetheless, in the decoder of sVCD, we simplify the architecture by employing a causal and single-directional RNN, since it is able to handle the relatively simple decoding task while incurring a low computational complexity.
>
> [R1] Vaswani, Ashish, Noam Shazeer, Niki Parmar, Jakob Uszkoreit, Llion Jones, Aidan N. Gomez, Łukasz Kaiser, and Illia Polosukhin. "Attention is All You Need." In Proc. of NeurIPS, pp. 5998-6008. 2017.
> [R2] Devlin, Jacob, Ming-Wei Chang, Kenton Lee, and Kristina Toutanova. "BERT: Pre-training of Deep Bidirectional Transformers for Language Understanding." In NAACL-HLT (1): 4171-4186, 2019.
>
> W1-b) We are using training sequences of the same length in our experiment. However, the length of the sequence can be varied since the RNN-centric structure does not impose any specific constraint on input length. To demonstrate the effect of varying input data length during testing, we have added an experiment by changing the data length, and the results are shown in the appendix of the revision. It can be observed that the performance of sVCD is not sensitive to the length of input signal.
>
> W1-c) Please refer to our earlier response to W0 for the answer.
>
> W2) We are grateful to the reviewer for identifying an irrelevant reference in our manuscript. In the revision, we would like to clarify that EEG signals are indeed nonlinear superpositions of the electrical activities of the neurons [R3], and the nonlinearity is escalated by the nonlinear properties of the brain [R4] and the skull [R5].
>
> [R3] Steyrl, David, Reinhold Scherer, and Gernot R. Müller-Putz. "Random Forests for Feature Selection in Non-invasive Brain-Computer Interfacing." In International Workshop on Human-Computer Interaction and Knowledge Discovery in Complex, Unstructured, Big Data, pp. 207-216. Springer, Berlin, Heidelberg, 2013.
> [R4] Nunez, Paul L., and Ramesh Srinivasan. Electric Fields of the Brain: The Neurophysics of EEG. Oxford University Press, USA, 2006.
> [R5] Zhang, Jinhua, Jiongjian Wei, Baozeng Wang, Jun Hong, and Jing Wang. "Nonlinear EEG Decoding based on a Particle Filter Model." BioMed Research International 2014 (2014).
>
> W3) Our interpretation of the reviewer’s concern is the following: what if we construct a new baseline without VI, but with a simple regularization on the L_2 norm of the mean. Based on this interpretation, we have conducted new ablation study in Section 5.4 to take this new baseline into account.

---

> > ### Comment · Reviewer_epBn · 2021-11-22
> > **still concerned**
> >
> > While there are useful contributions in the paper and I appreciate the revision, the text regarding the presentation of them method as a separation algorithm is  misleading.
> >
> > The paper initially discusses 'blind source extraction' in the context of 'blind source separation', but then switches to a supervised, non-linear, multiple-input single output regression task. Blind source separation is very different than the proposed approach and using these methods as comparisons is a straw man. At a minimum the paper should compare performance versus a linear model with supervision. Additionally, the paper should compare versus a non-linear but instantaneous regression model (not sequence to sequence).
> >
> > I thank the authors for the additional tests of varying sequence length, but I was expecting them to perform tests all the way down to single time step predictions and see where the prediction breaks. What is the minimal amount of a sequence is needed to begin to estimate the source of interest?
> >
> > For the supposed nonlinearity in EEG, I'd like to remind the authors that passive physical electrical systems are linear. For instance the skull and scalp, create a linear filtering effect on the brain's electrical fields. Clearly the central nervous system itself is highly non-linear. The neural activity at the macroscopic level is measured by the electrical fields created by 'sources' corresponding to the coordinated activity of multitudes of neurons. This generative process is no doubt non-linear with respect to the past and any stimuli. Nonetheless, the system describing the superposition and filtering of these electrical fields upon the electrodes is linear. Furthermore, just because a multiple input single output system is causal and linear it doesn't mean that the best inverse system would be linear. Thus, the paper's methodology does not require the system that created the mixture to be non-linear to require a non-linear demixing. Sequence to sequence can still be useful for demixing and deconvolving linear systems.

---

> > > ### Author Response · Authors · 2021-11-23
> > > **Response to Reviewer epBn**
> > >
> > > 1) We fully agree with the reviewer that only comparing with ICA might not be very fair. Therefore, we have added TasNet [R1] as a supervised signal separation baseline for comparison in our previous revision (please refer to Section 3.5 and Figures 4&5 for details). We know it is not a linear separation model as requested by the reviewer, but we guess it might not matter here, as TasNet (as speech separation model) should work on both linear and nonlinear mixtures.
> > >
> > > [R1] Luo, Yi, and Nima Mesgarani. "Conv-TasNet: Surpassing Ideal Time–frequency Magnitude Masking for Speech Separation." IEEE/ACM Transactions on Audio, Speech, and Language Processing 27, no. 8 (2019): 1256-1266.
> > >
> > > As for the “non-linear but instantaneous regression mode” asked by the reviewer, we thought about it earlier but we decided not to perform the comparison due to some well-known reasons: in order to perform conventional non-linear regression, it is crucial to select a proper kernel, which is similar to the handpicked features used in conventional machine learning. Nonetheless, this method is not sufficiently adaptive to all sorts of problems, and it is indeed for such reasons that we start to adopt deep learning method so as to obtain features/kernels in a data driven manner. In order to completely response to the reviewer, we have performed both linear and non-linear regressions for the instantaneous separation purpose, but we have put them in Appendix B (instead of reporting them in the main texts of the paper), as the results are far from usable due to the omission of temporal correlations.
> > >
> > > 2) We have added experiment results in appendix A to show the minimal amount of a sequence needed by sVCD to estimate the source of interest. Essentially, we believe that each interested source signal has distinct time-correlation features, and the sequence length needs to be long enough to contain these features. Therefore, we often take a sufficiently long sequence as it is hard to judge the minimum yet sufficient length a priori.
> > >
> > > 3) We agree with the reviewer that sVCD should work for both linear and non-linear mixtures, and a linear problem could be even easier to solve. However, we emphasize the non-linearity in our paper because it is a much harder problem (virtual unsolved) and hence the target of our sVCD.
> > >
> > > As for whether the human brain is linear or not, we are certainly not experts in those fields. Nonetheless, as suggested by the references provided in our previous response ([R3, R4, R5]), the very complex interconnections of the neuron, abundant soft tissues, and the irregularities of the skull all exhibit nonlinear electrical properties. We also believe that it might not be proper to model the human brain as a “passive physical electrical system”, since the brain is neither passive nor a simple object that can be abstracted as an electronic circuit. Well, again, we are not experts in neuroscience, and the aforementioned arguments all came from literature read and learned by us. Anyway, as pointed out by the reviewer, regardless of the signal mixture being linear or not, our results on sVCD show that it properly handles the problem of extracting EOG from EEG-EOG mixture.

---

> > > > ### Comment · Reviewer_epBn · 2021-11-28
> > > > **not claiming that the brain is linear**
> > > >
> > > > Sorry for the confusion, but I am not claiming the brain is linear. I'm say the effects of connective tissue and the skull can be described as a linear system acting on the brains electrical activity before it is recorded at the electrodes. I would encourage the authors to carefully re-read the original references to not misrepresent the information the information that they state.

---

> > > > > ### Author Response · Authors · 2021-11-30
> > > > > **We'd further emphasize on the nonlinear mixture nature of EEG and EOG**
> > > > >
> > > > > We are grateful to the reviewer for paying efforts on guaranteeing the correctness of our statements. Overall, our belief is that it is necessary to treat the mixing of EEG and EOG as nonlinear. We have identified a few other publications in the literature [R6, R7, R8]; they all claim that nonlinear mixing models should be better suitable to deal with EEG signals.
> > > > >
> > > > > [R6] Grilo, Marcelo, Layse Ribeiro, Caroline Moraes, Carlos Melo, Denis Fantinato, Leonardo Sampaio, Aline Neves, and Rodrigo Ramos. "Artifact Removal in EEG based Emotional Signals through Linear and Nonlinear Methods." In 2019 E-Health and Bioengineering Conference (EHB), pp. 1-4. IEEE, 2019.
> > > > > [R7] Molla, Md KI, Toshihisa Tanaka, Tomasz M. Rutkowski, and Andrzej Cichocki. "Separation of EOG Artifacts from EEG Signals using Bivariate EMD." In 2010 IEEE International Conference on Acoustics, Speech and Signal Processing, pp. 562-565. IEEE, 2010.
> > > > > [R8] Oveisi, Farid. "EEG Signal Classification using Nonlinear Independent Component Analysis." In 2009 IEEE International Conference on Acoustics, Speech and Signal Processing, pp. 361-364. IEEE, 2009.

---

> > > ### Author Response · Authors · 2021-11-26
> > > **Any further concern on reggresion?**
> > >
> > > Dear Reviewer,
> > >
> > > We have carefully responded to your earlier comments on missing comparison results with instantaneous regression model in our latest revision. We are wondering if they have addressed your concern? We would greatly appreciate if you could share with us your further opinion on our paper taking these newly added experiments into account.
> > >
> > > Thanks a lot!
> > >
> > > Best,
> > > Authors of Paper2186

---

> > > > ### Comment · Reviewer_epBn · 2021-11-28
> > > > **more regression would be great**
> > > >
> > > > Thank you for the Appendix B with the instantaneous regression. This shows how much the multivariate/spatial patterns can be used to predict the output.
> > > >
> > > > The next set of baselines I would like to see would take into account some time window to predict the regression target. For the linear model this would be known as the Weiner filter, or the stochastic gradient descent version: least-mean square (LMS) algorithm from Widrow and Hoff.
> > > >
> > > > For the non-linear case, I'd encourage the authors to go beyond kernel machines to also use neural networks (tapped delay line). There is a long history of neural networks for time series prediction. For kernel machines, there is the kernel version of LMS (KLMS)
> > > >
> > > > W. Liu, P. P. Pokharel and J. C. Principe, "The Kernel Least-Mean-Square Algorithm," in IEEE Transactions on Signal Processing, 2008, doi: 10.1109/TSP.2007.907881.
> > > >
> > > > I agree that kernel choice is difficult, and thus a neural network may be more appropriate. I also acknowledge that a tapped delay needs to select an appropriate window length a priori (or use a recursive neural network).  This points to the potential impact of the paper. Nonetheless, my concern is that these baselines are important to contextualize the solution. The reason they were left out is that the paper confounds blind source separation with non-linear (multiple input-single output) system identification.

---

> > > > > ### Author Response · Authors · 2021-11-30
> > > > > **more results have been obtained**
> > > > >
> > > > > We have conducted a new set of experiments by taking into account a time window when performing regression. To be specific, we employed both the KLMS and SVR algorithms, and the time window is set to 10 and 20. As we have passed the deadline for updating the paper, we may only orally explain the results here. Overall, the results are similar in median to those of the nonlinear regression (reported earlier in Figure 8 of the Appendix B), except that the standard deviations of the cosine similarity distribution become slightly smaller, since a multi-step regression utilizes more data hence resulting in better robustness. Based on these results, we admit that our earlier statement “the results (of instantaneous regression method) are far from usable due to the omission of temporal correlations” in the rebuttal is not totally correct, as temporal correlation is not the dominating reason for recovering qualified waveform; the versatile function approximation capabilities of deep neural network along with the variational inference module may have been the major reason for the success of sVCD, as also acknowledged by the reviewer.
> > > > >
> > > > > We have also tested a tapped-delay net implemented in Matlab (https://www.mathworks.com/help/deeplearning/ref/timedelaynet.html;jsessionid=9be72bb7d9c66f90092ffab6809f), due to limited time. As expected, its performance is inferior to that of Conv-TasNet (already compared in our earlier revision); this difference is rather intuitive as the latter is particularly designed for processing sequence data.

---

> ### Author Response · Authors · 2021-11-15
> **Response to Reviewer epBn (Cont’d)**
>
> We would like to thank the reviewer for constructive feedbacks and valuable comments. We address your comments about the minor points below.
>
> We have revised our manuscript by correcting typos in the figure and text. In particular, we have rephrased the sentence after Equation 1 to “the KL divergence can be minimized by maximizing the variational lower bound”.
>
> We keep the subscript notation 1: T introduced in Section 4.1 (top of page 5) as it is as the notation follows the convention in [R6]. The softmax notation in Equation 3 works indeed as suggested by the reviewer; it follows the convention in [R1].
>
> [R6] Marco Fraccaro, Søren Kaae Sønderby, Ulrich Paquet, and Ole Winther. "Sequential Neural Models with Stochastic Layers". In Proc. of NeurIPS, pp. 2207–2215. 2016.

---

### Official Review · Reviewer_MyNd · 2021-11-02

**Correctness:** 4
**Technical Novelty And Significance:** 4
**Empirical Novelty And Significance:** 3
**Recommendation:** 8
**Confidence:** 4

**Main Review:**

Strengths
1] Far above average clarity and overall writing quality and completeness
2] In section 5.4, the Ablation Study explores limits of the Gaussian assumption

Suggestions

1] "source" <-- "sour" at the top of page 2
2] Try comparing against more advanced technique (whther another deep learning model, or an algorithm that has been tuned to the specific application)

**Summary Of The Paper:**

Problem: blind source separation of all sources in a nonlinear mixture is very difficult, and potentially wasteful if you only need to extract one component
Solution: focus on extracting only the component that you need, by training a Variational encoder/decoder architecture (named a variational component decoder) to reconstruct only the component you wish to extract.

Additional contribution:
* lower bound of variation for the new architecture is proved

**Summary Of The Review:**

The authors make a strong theoretical justification of the model approach to source extraction from time series. The writing is exceptionally clear and informative and the experimental design is exceptionally sound. I scored 3/4  Empirical significance because the method is only compared to ICA, rather than (1) another more comparable deep model, or (2) state-of-the-art method in the specific tasks you are performing on. On the other hand I greatly appreciate that the authors explore the bound of their underlying Gaussian assumption in Sec 5.4. The technical novelty and significance is more exceptional due to the provided bound justifying the approach. Overall I think this is a solid "accept", also because the authors did something that worked and can explain why.

---

> ### Author Response · Authors · 2021-11-15
> **Response to Reviewer MyNd**
>
> We would like to thank the reviewer for appreciating the contribution of our work. In response to the comments on lack of comparison with a comparable deep model, we have added experiments in Section 5.3 to compare with a well-cited speech separation network [R1], which assumes linear mixtures. To best of our knowledge, there is no existing approach for extracting a component from a nonlinear mixture, existing solutions are all based on linear approaches.
>
> [R1] Luo, Yi, and Nima Mesgarani. "Conv-TasNet: Surpassing Ideal Time–frequency Magnitude Masking for Speech Separation." IEEE/ACM Transactions on Audio, Speech, and Language Processing 27, no. 8 (2019): 1256-1266.

---

### Official Review · Reviewer_LaJi · 2021-11-03

**Correctness:** 4
**Technical Novelty And Significance:** 2
**Empirical Novelty And Significance:** 3
**Recommendation:** 8
**Confidence:** 2

**Main Review:**

There is no obvious error I can see in the theory or experiments of this paper. Although I was not able to check the math carefully. The results seem sufficient to back the claims made by the authors. Although the techniques themselves are not novel, the application of these techniques to solve the non-linear Blind Source Separation problem is somewhat novel. Blind Source Separation is a ubiquitous problem and even though this work concerns itself just with one restricted version of the problem, where only one source is extracted, the applicability of this work is potentially quite significant. The results presented by the authors are satisfactory and show convincingly that the presented method is superior to other state-of-the-art techniques. I think the paper is good for acceptance, but I think it would be interesting if the authors could also relate their work with Sparse Coding and not just non-linear ICA.

**Summary Of The Paper:**

Authors present a novel approach that uses deep learning to solve non-linear Blind Source Separation problems. More specifically, their approach combines Seq2Seq and variational inference to extract one source of interest out of the nonlinear mixture. The generative model incorporates the prior beliefs about the source to be extracted.

**Summary Of The Review:**

The paper introduces a novel application of Seq2Seq and variational inference and the results show that the method's performance is superior to other state-of-the-art methods for a restricted version of Blind Source Separation where only one source is extracted from a non-linear mixture. My recommendation is to accept this paper for the conference.

---

> ### Author Response · Authors · 2021-11-15
> **Response to Reviewer LaJi**
>
> We would like to thank the reviewer for appreciating the contribution of our work. As for the relation with Sparse Coding, our understanding is that all encoder/feature extractor networks aim toward a sparse representation. However, our sVCD innovates in two aspects. On one side, we apply variational inference to regularize this representation so as to achieve generalizability. On the other side, this encoded (sparse) representation is for the extracted component (guided by the supervised learning enforced by the decoder) out of a nonlinear mixture. To the best of our knowledge, conventional Sparse Coding [R1], aims to find a sparse representation in the form of a linear combination of the input.
>
> [R1] Lee, Honglak, Alexis Battle, Rajat Raina, and Andrew Y. Ng. "Efficient Sparse Coding Algorithms." In Prof. of NeurIPS, pp. 801-808. 2007.

---

### Decision · Program_Chairs · 2022-01-20

**Decision:**

Reject

**Comment:**

This work has generated a lot of discussion between authors and reviewers and among reviewers.
Overall it is reported that the results on EEG are not conclusive and directly relevant for this field.
Besides the theoretical contribution is not reported as a strong point of the work and the
comparison with alternative baseline methods is judged too limited.

For all these reasons the paper cannot be endorsed for publication at ICLR this year.